

# Multi-Model Comparison in the Impact of Lateral Boundary Conditions on Simulated Surface Ozone across the United States Using Chemically Inert Tracers

Peng Liu[1], Christian Hogrefe[2], Ulas Im[3], Jesper H. Christensen[3], Johannes Bieser[4], Uarporn Nopmongcol[5], Greg Yarwood[5], Rohit Mathur[2], Shawn Roselle[2], Tanya Spero[2]

[1] NRC Research Associate, in the National Exposure Research Laboratory, U.S. Environmental Protection Agency, Research Triangle Park, NC, 27711, USA
[2] National Exposure Research Laboratory, U.S. Environmental Protection Agency, Research Triangle Park, NC, 27711, USA
[3] Aarhus University, Department of Environmental Science, Frederiksborgvej 399, 4000, Roskilde, Denmark
[4] Helmholtz-Zentrum Geesthacht, Institute of Coastal Research, Max-Planck-str. 1 21502 Geesthacht, Germany
[5] Ramboll, 773 San Marin Drive, Suite 2115, Novato, CA 94945, USA

*Correspondence to*: Peng Liu (liu.peng@epa.gov)

**Abstract.** This study represents an inter- comparison of four regional-scale air quality simulations, focused on understanding similarities and differences in the simulated impact of ozone lateral boundary conditions (LBCs) on the ground-level ozone predictions across the U.S. The chemically inert tracers were implemented in the simulations as a diagnostic tool to understand the similarities and differences between models at process level. For all simulations, three chemically inert tracers (BC1 BC2 and BC3) are used to track the impact of ozone specified at different altitudes along the lateral boundaries of the modeling domain encompassing the contiguous U.S. The altitude ranges specified for BC1, BC2, and BC3 broadly represent the planetary boundary layer (PBL), free troposphere, and upper troposphere-lower stratosphere, respectively.

The four simulations, namely WRF/CMAQ, WRF/CAMx, WRF/DEHM and COSMO-CLM/CMAQ, can have considerable differences in the simulated inert tracers at surface, indicating their different estimates in the impact of lateral boundary on surface ozone within the U.S. due to the physical processes alone in chemical transport models. WRF/CMAQ is used as a base case, and the differences between WRF/CMAQ and the other three models are examined, respectively. The model pair of COSMO-CLM/CMAQ and WRF/CMAQ shows the smallest differences in inert tracers, with the difference in BCT (sum of BC1, BC2 and BC3) peak in winter to be 1.6 ppb averaged across all sites. The model pair of WRF/DEHM and WRF/CMAQ shows the largest differences, with difference in BCT peak in summer to be 8.1 ppb averaged across all sites. Furthermore, the model differences in inert tracers are discussed with respect to the physical processes that inert tracers undergo. It is found that the process of vertical turbulent mixing between the PBL and the free troposphere is the main cause of the model differences in the simulated inert tracers, especially the relative contributions of BC1 and BC2 to the total inert tracers, in most seasons and regions of the U.S., although the processes of sub-grid cloud mixing and dry deposition can also be important drivers for specific regions and seasons.





## 1 Introduction

Studies based on chemical transport models (CTMs) have shown that air quality in the U.S. can be considerably influenced by pollutants beyond the U.S. boundaries, such as through intercontinental transport, and through stratosphere-to-troposphere exchange (Zhang et al., 2011; Lin et al., 2012; Nopmongcol et al., 2016; Langford et al., 2017; Lin et al., 2017; Hogrefe et al., 2017; Mathur et al., 2017). Similar findings have also been reported based on routine observations and field campaign measurements (e.g. Cooper et al., 2012; Gratz et al., 2015; Langford et al., 2015), especially at the rural and elevated locations in the western U.S. Recent revisions to the National Ambient Air Quality Standards (NAAQS) for ground-based ozone (Federal Register, 2015), which lowered both the primary (health-based) and secondary (welfare-based) standards from 75 ppb to 70 ppb, have placed increasing emphasis on the need to characterize the contributions and the uncertainties of ozone imported from outside the U.S.

To quantify the impact of emissions and pollutants from outside the region of interest, different approaches have been implemented in CTMs and broadly fall into two categories. The first category is called "source sensitivity approach", such as "brute-force" method and the decoupled direct method in 3-dimensions (DDM-3D) (Dunker, 1984; Yang et al., 1997; Dunker et al., 2002). The second category is called "source apportionment approach". For example, pollutants from different sources can be tracked directly by adding chemically reactive tracers to CTMs. Based on the approaches above, a variety of model-specific tools have been developed and enhanced for different CTMs, such as DDM-3D for the Community Multiscale Air Quality (CMAQ) model (Cohan et al., 2005) and for the Comprehensive Air quality Model with extentions (CAMx) (Yarwood et al., 1996; Dunker et al., 2002), respectively, the Integrated Source Apportionment Method (ISAM) for CMAQ (Kwok et al.,2015), the Ozone Source Apportionment Technology (OSAT) for CAMx (ENVIRON, 2015), the adjoint model for GEOS-Chem (Henze et al., 2007), and a tagging scheme for ozone production from NO sources in the Model for Ozone and Related chemical Tracers (MOZART-4) (Emmons et al., 2012). With the help of these tools, the impact of ozone from outside the U.S. on the surface ozone within the U.S. has been quantitatively estimated (e.g. Baker et al., 2015; Dolwick et al., 2015; Nopmongcol et al., 2017).

Any single model realization of CTMs with the implementation of these approaches, however, may not be enough to describe the complete picture of interest due to the uncertainties in air quality modeling, which stem from the model inputs (e.g. Hanna et al., 2001) and the CTMs themselves (as reviewed by Russell and Dennis, 2000). Multi-model ensembles can help with uncertainty quantification in air quality modeling (Mallet and Sportisse, 2006; Vautard et al., 2006). In addition, the prediction by a multi-model ensemble may outperform any individual member (e.g. van Loon et al., 2007; Langner et al., 2012). Creating multi-model ensembles is however resource intensive and often reliant on extensive collaborations across multiple participating organizations. For example, about 40 institutions from the U.S. and Europe have participated in the three phases of the Air Quality Model Evaluation International Initiative (AQMEII) (Galmarini et al. 2012, 2015, 2017), enabling the inter-comparison across a variety of CTMs and the examination of the multi-model ensembles for both the U.S. and the Europe. Substantial effort in the past multi-model studies has focused either on the operational evaluation (Dennis et



al., 2010) across models, in which modeled results are compared against corresponding measured data (e.g. Solazzo et al., 2012; Im et al, 2015), or on the generation and analysis of multi-model ensembles (e.g. Delle Monache and Stull, 2003; Mallet et al., 2009; Galmarini et al. 2013; Kioutsioukis et al., 2016). Only limited efforts have been devoted towards elucidating the reasons for the noted similarities/differences in surface ozone prediction among the models. Campbell et al.

(2015), for example, compared the indicators of ozone sensitivity across regional CTMs in Phase 2 of AQMEII. In Phase 3 of AQMEII (AQMEII3), Solazzo et al. (2017) examined how model errors in surface ozone prediction can be apportioned to different time scales and how the errors could be related to the chemical and physical processes in CTMs. Such process-level diagnostic inter-comparison of CTMs is important for several reasons. First, comparison in fundamental processes can help to achieve the scientific advances in the representations of the physical and chemical processes in CTMs, so that model

prediction from a single CTM can be improved, as reviewed by Zhang et al. (2012). Second, process-level multi-model comparisons can help to guide the research directions to reduce model uncertainty by identifying the major process(es) that contributes to the variability across models. Last, a better understanding of the model similarities and differences at the process level could improve multi-model ensembles by increasing the independence of ensemble members.

This study, performed under AQMEII3, aims to investigate how different representations of physical processes and different

model configurations in CTMs may lead to the differences in the estimated impact of ozone imported from outside the U.S. on the surface ozone within the U.S. To enable such diagnosis, all participating groups in AQMEII3 implemented chemically inert tracers to track the transport, scavenging and deposition of ozone imported from the lateral boundaries into the simulation domain, which covers the COntiguous United States (CONUS) and southern Canada (Figure 1). It should be noted that since chemical loss is not considered, the inert tracers should only be regarded as a diagnostic tool to help

understand the similarities and differences between models, rather than a tool for the long-range-transport attribution. The inert tracer method has its advantage in two aspects. First, compared with the "source sensitivity" and "source apportionment" approaches mentioned above, the method of inert tracers is much easier to be consistently implemented in different CTMs by all AQMEII3 participants. Second, since the representation of chemistry differs across the CTMs, the inert tracer method isolates the differences related to the representation of 3-D transport (advection, turbulent mixing and

cloud mixing) and the physical sinks from the differences in chemical processes in CTMs.

This paper is organized as follows. Section 2 describes the model configurations and how the chemically inert tracers are implemented. In section 3, the impact of lateral boundary ozone due to the representations of physical processes and the model configurations in CTMs is compared among models by season. The model differences are investigated and discussed with respect to the physical processes that inert tracers are involved. Section 4 discusses how the model differences noted in

section 3 may change if chemical loss of the ozone from lateral boundaries was also considered.



## 2 Methods

### 2.1 Model description

This study investigates simulations conducted by four different research institutes/groups from the U.S. and Europe, using state-of-the-art regional CTMs. Based on the combination of the models used for the meteorological driving fields and

CTMs, the four simulations are named as WRF/CMAQ, WRF/CAMx, COSMO-CLM/CMAQ, and WRF/DEHM in this study. Important model details are summarized in Table 1, with additional model features available in Solazzo et al. (2017). Two additional sensitivity simulations were also available, namely WRF/CMAQ_NODDEP and WRF/DEHM_NODDEP, in which all settings were the same as WRF/CMAQ and WRF/DEHM, respectively, except that the dry deposition of inert tracers was turned off. The simulation period consists of the entire year of 2010, which was determined by AQMEII3 based

on the data availability of emissions and observations. The four simulations share the same chemical boundary conditions derived from the Composition Integrated Forecasting System (C-IFS) global modeling system (Flemming et al., 2015) by the European Centre for Medium-Range Weather Forecast (ECMWF). The anthropogenic emissions were harmonized to be consistent across the models under the effort of AQMEII3. For WRF/DEHM and COSMO-CLM/CMAQ, the anthropogenic emissions were prepared on a gridded and monthly basis using HTAP_v2.2 database (Janssens-Maenhout et al, 2015); while

for the other simulations, the anthropogenic emissions were provided by U.S. EPA (Pouliot et al., 2015) on a gridded and hourly basis. The North American emissions in the HTAP_v2.2 database were compiled using the U.S. EPA emissions and therefore, the two sets of emission data in North America have the same annual total for each species in each sector. Non-anthropogenic emissions were estimated separately by each individual research group/institute. For example, biogenic emissions for WRF/CMAQ were estimated using the Biogenic Emissions Inventory System (BEIS) v3.14 (Schwede et al.,

2005; Pouliot et al., 2015) with USGS land use data, while the biogenic emissions for WRF/CAMx were determined by the Model of Emissions of Gases and Aerosols from Nature (MEGAN) v2.1 (Guenther and Wiedinmyer, 2007; Sakulyanontvittaya et al., 2008) with land use data from the North American Land Change (NALC) Monitoring System. A comprehensive description of the model activity and emissions related to AQMEII3 can be found in the technical note by Galmarini et al. (2017). In short, with the current modeling setup, the results from the four simulations mainly reflect the

variability due to different meteorological driving fields, model configurations and the representation of processes in CTMs, all of which are important components of the uncertainty in air quality modeling (Fox, 1984; Solazzo and Galmarini, 2016).

### 2.2 Chemically inert tracers

For each simulation, three chemically inert tracers were added specifically at the lateral boundaries to track ozone from outside of the modeling domain. The tracers undergo the same physical processes as ozone, including advection, diffusion,

cloud mixing/transport (if represented in CTMs, as summarized in Table 1), scavenging and deposition, with no emissions or chemical formation/destruction. The only exception is that WRF/CAMx does not include any wet deposition for the inert tracers (Nopmongcol et al., 2017). In all models, the deposition velocity of tracers was set to be the same as that of ozone.

The three tracers, representing the lateral boundary ozone from the planetary boundary layer (PBL), the free troposphere, and the upper troposphere-lower stratosphere respectively, are defined as follows: BC1 for vertical layers below 750 mb (~ 2.5 km); BC2 for layers between 750 mb (~2.5 km) and 250 mb (~10 km); and BC3 for layers above 250 mb. Initial conditions for all tracers were set to be zero and a ten-day spin-up period was used in the simulations. The lateral boundary conditions (LBCs) of the tracers were set to be the same values as the LBCs of ozone at the corresponding vertical layers, with zero values assigned in other layers. Therefore, these tracers are able to provide information on how specified ozone LBCs at different altitude ranges may eventually influence the simulated surface ozone in the U.S.

### 2.3 Data for analysis

As required by AQMEII3, the modeled hourly values for ozone, tracers and meteorological fields were uploaded to the ENSEMBLE data system developed by the EC-Joint Research Center (Galmarini et al., 2001) for the purpose of sharing results between AQMEII3 participants. For ozone, the inert boundary condition tracers, and other trace gases, AQMEII3 participants were required to provide the modeled hourly values at the surface at the locations where ozone monitors are available. The observation networks constituting this set of monitor locations include the Clean Air Status and Trends Network (CASTNET), the Air Quality System (AQS), the SEARCH network for the U.S. and the National Air Pollution Surveillance (NAPS) network for Canada. In the studied area, there are 278, 614 and 587 monitors for urban, rural and suburban locations respectively (Figure 1). All multi-model comparison in section 3 was conducted based on the surface data at these monitors, unless otherwise specified. The fact that only surface level concentrations were available for this study poses a limitation to disentangling the impact of lateral boundary ozone on surface ozone at process-level.

### 3 Results

For all simulations involved in this study, it has been found that the seasonally averaged contribution of BC3 to the total inert tracers (namely the sum of BC1, BC2 and BC3, hereafter referred to as BCT) is in general small at the surface (less than %5 on a seasonal average basis). Similar findings have also been reported by Nopmongcol et al. (2017). Furthermore, the differences in seasonally averaged BC3/BCT at the surface among models are also usually small (within ± 2.5%), compared with the differences in (BC1/BCT) and in (BC2/BCT). Therefore, only the results of BC1, BC2 and BCT will be shown in the subsequent analysis.

### 3.1 Multi-model comparison in the impact of lateral boundaries using inert tracers

The model results for BCT are examined first, as BCT represents the impact of lateral boundary ozone from all altitude ranges due to the physical processes in CTMs. WRF/CMAQ is used as a base case, and the differences in daily maximum 8-hour average (DM8A) BCT between WRF/CMAQ and the other 3 models are shown on a seasonal basis (Figure 3). For each model, DM8A BCT is calculated at the grid cells where the monitors are located as follows. First, for each day, the



eight-hour window when the modeled DM8A ozone occurs is found. Then the averaged mixing ratios of BCT during that eight-hour window are calculated using the modeled hourly data at surface, which are referred to as DM8A BCT. Finally, the DM8A BCT are averaged for each season.

Considerable differences are seen between the models both spatially and seasonally, indicating that significant model differences in the impact of lateral boundary ozone can result from the physical processes in CTMs alone. For example, BCT from WRF/CAMx can be higher than that from WRF/CMAQ by as much as about 5 ~ 7.5 ppb for some regions during winter and summer; BCT from WRF/DEHM can be lower than that from WRF/CMAQ by as much as about 12.5 ~ 17.5 ppb in the southeastern U.S. during summer.

Meanwhile, in some instances, similar results in DM8A BCT (differences smaller than 2.5 ppb) between models are also found. WRF/CMAQ and WRF/CAMx show similar BCT results during fall, as do WRF/CMAQ and COSMO-CLM/CMAQ in all seasons at most locations. The similar model results for BCT, however, do not necessarily indicate model similarity in the physical processes that inert tracers undergo, because the four simulations share the same chemical boundary conditions, including the LBCs for the inert tracers. Hence, it is needed to confirm that the inert tracers at the surface are primarily affected by the physical processes in the CTMs per se, instead of by the LBCs.

This issue is investigated by examining the relative contributions of BC1 and BC2 to BCT. If the inert tracers at the surface are determined by LBCs, little spatial variability would be expected in the relative contributions of tracers. For all simulations, it is found that the relative contributions of BC1 and BC2 vary significantly across the modeling domain. The ratio of (DM8A BC1) to (DM8A BCT) from WRF/CMAQ is shown as an example (Figure 4, first row). The ratio of (DM8A BC2) to (DM8A BCT) is not shown, as the ratio of (BC1+BC2) to BCT is almost 100%. The values of DM8A BC1 and DM8A BC2 are calculated following the same steps for DM8A BCT. It is also found that (DM8A BC1)/(DM8A BCT) along the western and northern U.S. boundaries is usually higher than the inner U.S., due to higher DM8A BC1 (Figure 4, second row) and lower DM8A BC2 (Figure 4, third row) along the two boundaries. This is because the western and northern boundaries of the simulation domain are the predominantly inflow boundaries of the modeling domain, especially in winter and spring. In addition, since the LBCs of BC2 are zero below 750 mbar by definition, grid cells close to these two boundaries tend to have higher DM8A BC1 and lower DM8A BC2 at the surface than those in the interior portions of the domain. In other words, the relative impact of LBCs and physical processes on inert tracers at the surface varies spatially, with larger impact of LBCs on the grid cells close to these two boundaries than the grid cells in the inner domain. Such features are also noticed in other models.

In short, the spatial distribution of the relative contributions of tracers clearly shows that the physical processes in the CTMs are involved in determining the inert tracers at the surface for the studied area. Hence, the model inter-comparison in inert tracers can be used to indicate similarities and differences between models regarding these physical processes.



### 3.2 Multi-model comparison in the impact of physical processes on inert tracers

A variety of physical processes are involved in determining the mixing ratio of inert tracers at the surface, including 3-D advection, turbulent mixing, sub-grid cloud mixing, and wet and dry deposition. The model differences and similarities in BCT for each model pair shown in the previous section will be explained and discussed in context of these physical

processes by comparing model predictions in BC1, BC2 and their relative contributions to BCT.

### 3.2.1 WRF/CMAQ

The results from WRF/CMAQ are first examined, since WRF/CMAQ is used as the base case for the model inter-comparison.

The most important feature for BCT in WRF/CMAQ is that for all seasons, higher DM8A BCT is seen in the mountain west
region than the eastern U.S. (Figure 3, first row) due to the higher BC2 in the west (Figure 4, third row). This spatial contrast in inert tracers is not surprising because the higher elevation in the mountain region tends to generate more mixing between the mid-troposphere and the PBL, resulting in higher BC2 at surface in the mountain west U.S.

It is also expected that stronger mixing between the mid-troposphere and the PBL in the mountain region helps to dilute BC1 at surface. However, compared with the eastern U.S., lower levels in BC1 is not found in the mountain west region except
for winter (Figure 4, second row). One possible explanation is that the impact from vertical mixing and the impact from LBCs cancel out in the mountain west region. Since we are examining surface concentrations and since the western boundary is predominantly an inflow boundary, it should be expected that BC1 in the west is higher than that in the east due to the impact of LBCs alone. Another possible explanation is that physical processes other than vertical mixing also play a role in determining tracers at surface, such as dry deposition, as a clear spatial contrast in the ozone dry deposition velocity
occurs between the western and the eastern U.S. (Figure 5, first row), especially in spring and summer. This is further illustrated through the sensitivity simulation WRF/CMAQ_NODDEP. In the absence of dry deposition, surface-level mixing ratios of BC1 and BC2 increase across the domain, especially during spring and summer (Figure 4), but the increase is not uniform across the domain. Without the impact of dry deposition, lower BC1 is seen in the mountain west region than the eastern U.S. for all seasons.
The results from WRF/CMAQ and WRF/CMAQ_NODDEP clearly show that vertical mixing and dry deposition interact to determine the spatial contrast in BCT at the surface. Due to the higher elevation in the mountain western U.S., BC1 in the west tends to be lower than the east through vertical transport. Meanwhile, more BC1 is removed from the atmosphere in the eastern U.S. than the mountain west region due to dry deposition, except for winter, when the impact of dry deposition is almost uniform spatially. As a result, BC1 levels at surface is relatively uniform across the interior of the U.S. compared
with BC2. For BC2, on the contrary, the difference between the mountain western and the eastern U.S. is magnified when the two processes come into play together.  In short, the differences in vertical transport (due to elevation difference) and dry



deposition between the eastern and the mountain western U.S. mainly contribute to the spatial contrast of inert tracers at surface.

Other physical processes may also play a role in determining the mixing ratio of inert tracers regionally during certain season. For example, DM8A BC2 in eastern U.S. from WRF/CMAQ_NODDEP (Figure 4) is comparable to that in the west

during summer, indicating that other physical process(es) may exist to enhance the mixing of BC2 from mid-troposphere downward into PBL in the eastern U.S., such as sub-grid cloud mixing. However, with the current data available, it is not possible to further diagnose the cause of the differences at the process level.

Last, results of (DM8A BC1/DM8A BCT) clearly show that throughout the interior U.S., BC2 plays a dominant role in BCT. Additionally, although dry deposition significantly changes the absolute mixing ratios of the simulated tracers, it changes

little in the relative contributions of the tracers (Figure 4), since the dry deposition flux is in general proportional to the absolute mixing ratio of the species.

### 3.2.2 WRF/CAMx vs WRF/CMAQ

This model pair has some important features in common for air quality modeling, which the other model pairs do not have. The two models used the same meteorological inputs for CTMs and were configured with the same horizontal resolution.

However, it is interesting to find that the differences in BCT at the surface for this model pair are still comparable to and sometimes even larger than those in the other model pairs (Figure 3).

Comparing WRF/CAMx with WRF/CMAQ, for all seasons, WRF/CAMx shows consistently lower DM8A BC1 and higher DM8A BC2 across the U.S., leading to lower (DM8A BC1/DM8A BCT) for all seasons (Figure 6) and higher DM8A BCT except for fall (Figure 3). In fall, the differences in BC1 and BC2 are comparable in magnitude with the differences in other

seasons, but tend to cancel out each other, leading to smaller difference in BCT. The results suggest that a systematic difference exist between the two models, and such systematic difference overwhelms other differences in the model configurations and the representations of physical processes, leading to consistent differences in the simulated tracers across season and space. The two simulations were driven by the same meteorological inputs and utilized the same horizontal resolution, so that the horizontal transport and vertical advection should not in principle be the dominant contributor to these

modeled differences in inert tracers.

As to the vertical turbulent mixing, WRF/CMAQ used the parameterization ACM2 (Pleim 2007) for vertical diffusion, while WRF/CAMx used "K-theory". Compared with ACM2, K-theory may be less efficient in the mixing of the convective boundary layer during the deep vertical convection (ENVIRON, 2015). During the neutral and stable conditions, both parameterizations are able to adequately characterize the vertical mixing (ENVIRON, 2015). Therefore, if the

parameterizations for vertical turbulent mixing dominate the differences in simulated inert tracers, larger differences are expected to exist during summer, when deep convection is most frequent, rather than the consistent differences across season.





Another important systematic difference for this model pair is the vertical grid resolution, which may result in significant difference in the vertical diffusion between the two models. Though the two models have similar vertical structure from surface to about 900 mbar (~ 1km), WRF/CAMx used coarser vertical resolution from 900 to 300 mbar (Figure 2), which may lead to higher vertical mixing between the PBL and the free troposphere and eventually tend to dilute the BC1 levels at

surface and entrain greater amounts of BC2 downward to the surface.

In addition, sub-grid cloud mixing is also different in the two models (Table 1). In general, clouds enhance the vertical mixing between PBL and free troposphere, thus decreasing BC1 and increasing BC2 within the PBL. Lacking the representation of this process, WRF/CAMx would be expected to have higher BC1 and lower BC2 than WRF/CMAQ if all other physical processes were similar. However, the results show the opposite, indicating that the processes other than sub-

grid cloud mixing dictate differences in tracers, though locally the impact of this process can be as important as the impact of vertical grid resolution. As an example, along the gulf coast, where convection is usually frequent and strong (especially during spring and summer), the differences in BC1 and BC2 between the two models (within ±2.5 ppb) are smaller than the inner regions (Figure 6). A possible explanation is that the impact of sub-grid cloud mixing and the impact of vertical turbulent mixing cancel out. Though WRF/CAMx tends to mix more BC2 downward into the PBL and mix more BC1 into

the free troposphere than WRF/CMAQ due to its coarser vertical grid resolution in the free troposphere, the differences in tracers between the two models decrease due to the process of sub-grid cloud mixing in WRF/CMAQ along the gulf coast.

As to the physical sinks for inert tracers, WRF/CAMx did not include wet deposition of the inert tracers, the concentrations of tracers are expected to be higher than those in WRF/CMAQ. However, the opposite was found for BC1. Furthermore, the impact of wet deposition on pollutants usually shows a strong seasonality, with peak in summer or spring, but no seasonal

variation was found in the differences between the two models. Hence, wet deposition is unlikely to dominate the systematic model differences in inert tracers either. Last, the impact of dry deposition is investigated. Significant difference is found regarding the dry deposition velocity of ozone for this model pair (Figure 5, second row), resulting from the differences in the dry deposition scheme (Table 1) and the land use data. However, the difference in dry deposition velocity is not uniform spatially and temporally, which is opposite to the uniform spatial and temporal distributions in the differences in inert

tracers. Furthermore, the differences in BC1 and BC2 are opposite in sign (Figure 6), and thus, it is impossible that the difference in dry deposition between the two models dominates the differences in BC1 and BC2 at the same time. Therefore, we believe that dry deposition process does not dominate the systematic model differences in inert tracers. In addition, as shown in the comparison between WRF/CMAQ and WRF/CMAQ_NODDEP (Figure 4), dry deposition changes little on the relative contributions of inert tracers. Therefore, the significant difference in (DM8A BC1/DM8A BCT) for this model pair

does not result from the process of dry deposition.

To summarize, the model difference in vertical mixing due to the different vertical grid resolutions likely overwhelms other differences in the model configurations and the representations of physical processes throughout all seasons across the entire U.S. domain, though other process(es), such as sub-grid cloud mixing, can also be important locally during specific season.



### 3.2.3 WRF/DEHM vs WRF/CMAQ

Comparing WRF/DEHM with WRF/CMAQ, the difference in BCT (Figure 3) is not dominated by the difference in BC2 alone, as found in WRF/CAMx and WRF/CMAQ. Instead, whether BC1 or BC2 dominates the differences in BCT depends on season and location (Figure 7). In addition, considerable spatial variability is also found in the difference in BC1, BC2 and (DM8A BC1/DM8A BCT) (Figure 7). For example, large positive and negative differences (more than 15%) in (DM8A BC1/DM8A BCT) coexist within the U.S. except for winter, and this feature is not seen in the other model pairs. One possible explanation is that the differences in the modeled inert tracers for this model pair were determined by multiple processes at the same time. It is also possible that certain model representation of physical processes or model configuration mainly contributes to the differences in inert tracers, however, the impact of this factor on inert tracers changes spatially and temporally. Our analysis shows that the latter is likely to be the case.

The impact of dry deposition on the differences in simulated inert tracers is first investigated using WRF/CMAQ_NODDEP and WRF/DEHM_NODDEP sensitivity simulations. Without dry deposition, the model difference in DM8A BCT significantly decreases (Figure 3, last row) across all seasons, especially for spring and summer. This is because the differences in BC1 and BC2 are usually similar in magnitude but opposite in sign across most of the U.S. (Figure 7), leading to smaller differences in BCT. And by comparing the (DM8A BC1/DM8A BCT) from the model pairs with and without dry deposition (Figure 7), it is reconfirmed that the process of dry deposition has little impact on the relative contributions of inert tracers at surface. Then, the model pair of WRF/CMAQ_NODDEP and WRF/DEHM_NODDEP is used to investigate the impact of other physical processes on the differences in inert tracers. The impact of wet deposition on the differences in inert tracers for this model pair is believed to be negligible for two reasons. First, the differences in BC1 and BC2 between WRF/CMAQ_NODDEP and WRF/DEHM_NODDEP are found to be in opposite sign across the majority of U.S. Hence, this physical sink is unlikely to contribute and dominate the differences in BC1 and BC2 simultaneously. Secondly, the model difference in precipitation is in general small, except that significant higher precipitation is found in the southeastern U.S. in WRF/DEHM during summer (not shown). Though BC2 in WRF/DEHM_NODDEP is lower than that in WRF/CMAQ_NODDEP for this region in summer, which could result from wet deposition, the higher BC1 in WRF/DEHM_NODDEP is not consistent with its higher precipitation.

Given that the physical sinks are likely to play a minor role in the differences in the simulated inert tracers between WRF/CMAQ_NODDEP and WRF/DEHM_NODDEP, the representation of 3-D transport (advection, turbulent mixing and cloud mixing) should primarily contribute to the differences in two ways, namely (1) the meteorological inputs of CTMs to drive 3D advection and (2) the parameterizations for turbulent mixing and cloud mixing in CTMs.

WRF/DEHM was run at a much coarser horizontal resolution than WRF/CMAQ (Table1) and the meteorological inputs for the two models were generated from different global climate models, so that considerable differences in the meteorological inputs for CTMs could exist between the two models. The model differences in horizontal transport has been suggested by a previous study in AQMEII3 (Solazzo et al.,2017), in which the surface wind is compared over southeastern, northeastern



U.S. and the region of California with noticeable differences found for all seasons. However, the feature of the opposite signs for the differences in BC1 and BC2 in most regions of the U.S. between WRF/CMAQ_NODDEP and WRF/DEHM_NODDEP (Figure 7) suggests that the model differences in horizontal transport, especially within PBL, should not be the dominant reason behind the model difference in inert tracers. A possible argument for the opposite signs in

the differences in BC1 and BC2 is that the model differences in horizontal transport change with altitude range. For example, for spring, the lower BC1 in the central U.S. (Figure 7, fifth row) could result from the horizontal transport within the PBL; while the higher BC2 in this region (Figure 7, sixth row) could be initiated by higher BC2 in the free troposphere due to the opposite difference in horizontal transport in the free troposphere. Though it is not possible for this study to further investigate the model differences aloft due to the limitation of the availability of surface data only, we do not think the model

differences in the horizontal transport of inert tracers in the free troposphere could be the main reason for the differences in the modeled inert tracers at the surface. This is because the mixing ratios of BC2 in the free troposphere should be in general uniform across the modeling domain, especially on a seasonal basis.

Therefore, we believe that the differences in vertical transport between the two models are very likely to be mainly responsible for the differences in inert tracers at the surface. With limited meteorological data at the surface, the difference in

PBL height during daytime between the two models is used to indicate the potential model differences in vertical turbulent mixing. It should be noted that the definition of PBL height depends on the parameterizations of PBL (as summarized by Banks et al., 2016). The meteorological fields of the two models were generated using different PBL schemes. For WRF/DEHM, MYJ scheme (Mellor and Yamada, 1974, 1982; Janjić, 2002) was used, in which PBL height is defined by a prescribed threshold of turbulent kinetic energy. For WRF/CMAQ, ACM2 scheme (Pleim, 2007) was used, in which PBL

height is determined by the Richardson number calculated above neutral buoyancy level. Therefore, the difference in PBL height between the two models can partially stem from the different definitions of PBL height, instead of resulting from different vertical turbulent mixing alone. However, compared with other meteorological data available, such as wind at 10 m height and temperature at 2 m height, PBL height is still a better indicator for vertical turbulent mixing.

It is found that the differences in inert traces in some regions are consistent with the differences in PBL. For example, in

winter, WRF/DEHM shows higher PBL than WRF/CMAQ in the eastern U.S. (Figure 8, first row), which may explain the lower BC1 and higher BC2 at surface (Figure 7) in this region. In summer, WRF/DEHM shows much lower PBL than WRF/CMAQ in the southeastern U.S., which may explain the higher BC1 and lower BC2 in this region. In spring, however, the differences in inert tracers over this region do not match the difference in PBL height, especially in the area of Florida. The results suggest that the model differences in other physical process(es), such as sub-grid cloud mixing could also

influence the vertical mixing of inert tracers, so that the model differences in vertical turbulent mixing fail to dominate the model difference in inert tracers over this region. Unfortunately, with the current data available, it is impossible to further separate the impact of these processes.

It is interesting to note that the model differences in vertical transport mainly contribute to the differences in the modeled inert tracers for the model pair of WRF/CMAQ_NODDEP-WRF/DEHM_NODDEP and the model pair of WRF/CMAQ-



WRF/CAMx as well. However, consistent differences in inert tracers are only found in WRF/CMAQ-WRF/CAMx, but not in WRF/CMAQ_NODDEP-WRF/DEHM_NODDEP. This is because for WRF/CMAQ-WRF/CAMx, the coarser vertical resolution in the free troposphere in WRF/CAMx always tends to dilute the BC1 levels at surface and entrain greater amounts of BC2 downward to the surface, which is affected little by space and time. However, for WRF/CMAQ-

WRF/DEHM, the mixing between PBL and the free troposphere in WRF/DEHM can be stronger or weaker than that in WRF/CMAQ depending on the meteorological conditions and the responses of the parameterization of turbulent mixing to the meteorological conditions. As a results, though the differences in BC1 and BC2 between WRF/CMAQ_NODDEP-WRF/DEHM_NODDEP are usually in opposite sign, as what is also found in the model pair of WRF/CMAQ-WRF/CAMx, the differences show large spatial and temporal variability.

**3.2.4 COSMO-CLM/CMAQ vs WRF/CMAQ**

Comparing COSMO-CLM/CMAQ with WRF/CMAQ, it is found that this model pair has smaller differences in DM8A BCT than the other model pairs. Furthermore, the two models also show similar BC1 and BC2, with differences less than 2.5 ppb for most of the locations. The model differences in the relative contributions of BC1 and BC2 (Figure 9) are also much smaller than other model pairs, except over the southeastern U.S. during summer.

Given that the two models are driven by the same global meteorology and nudging is also applied above the PBL, the model differences in horizontal transport and vertical advection are expected to be small, especially at the spatial and temporal scales considered in this study. For dry deposition, it has been shown that this process barely changes the relative contributions of inert tracers. For wet deposition, the model difference in precipitation is also small (not shown). For sub-grid cloud mixing, its impact, if any, is not expected to be important across the whole U.S. for all seasons. Therefore, the

similar model performance for this model pair is mainly due to the similar representation of turbulent vertical mixing. Similar to the previous model pair, the PBL height in daytime is used to suggest the potential model differences in vertical turbulent mixing. For COSMO-CLM/CMAQ, an extended MYJ scheme was used (Doms et al., 2011). As mentioned before, the PBL height can only partially suggest the potential differences in vertical turbulent mixing, as PBL height is defined in different ways in different parameterizations.

It is interesting to find that COSMO-CLM/CMAQ has consistently higher PBL than WRF/CMAQ (Figure 8, second row), suggesting that more BC2 aloft is expected to be mixed downward into PBL and more BC1 to be diluted in PBL. However, the stronger vertical turbulent mixing in COSMO-CLM/CMAQ is not strong enough to overwhelm the other differences for this model pair, so that COSMO-CLM/CMAQ does not have consistent higher DM8A BC2 and lower DM8A BC1 than WRF/CMAQ (Figure 9). Such results are in opposite to the results seen for the model pair of WRF/CAMx and

WRF/CMAQ, in which stronger vertical mixing in WRF/CAMx has a dominant impact on the model difference in inert tracers. The stronger vertical mixing in COSMO-CLM/CMAQ is mainly due to different meteorological fields, because COSMO-CLM/CMAQ and WRF/CMAQ have very similar vertical structure from surface to until 400 mbar (Figure 2). The stronger mixing in WRF/CAMx is due to its coarser vertical structure, especially from 900 to 300 mbar. The contrasting




results of the differences in inert tracers in these two model pairs further suggest the important role that vertical grid resolution (both within the PBL and in the mid-troposphere) plays on the eventual simulated boundary tracer levels at the surface. In addition, though different PBL schemes were used by COSMO-CLM/CMAQ and WRF/CMAQ when the meteorological fields were generated, the same vertical turbulent mixing scheme was used for the CTMs, which may also

contribute to the small model differences in the inert tracers.

During the summer over the southeastern U.S., the model differences in other processes may become important as well. COSMO-CLM/CMAQ simulates much less precipitation (sum of convective and non-convective rain) than WRF/CMAQ, which could lead to less scavenging and higher inert tracer levels. Meanwhile, this region typically experiences frequent and strong convective activity during summertime. Less precipitation in COSMO-CLM/CMAQ could indicate weaker

convection and weaker sub-grid cloud mixing, leading to less BC2 brought downward from the mid-troposphere to PBL. The impact of less sub-grid cloud mixing may be stronger than the impact of less scavenging and finally result in lower BC2 in the COSMO-CLM/CMAQ simulations.

## 4 Summary and Discussions

In summary, the inert tracer method is a good balance of the needs for simplicity and the ability to provide diagnostic

information for the model inter-comparison with respect to the physical processes that the tracers undergo. The four simulations show considerable differences in their estimates on the impact of lateral boundary ozone due to the model configurations and the representations of physical processes, though the magnitude of the differences varies from season to season, and from model to model. For all the processes that the inert tracers undergo, vertical turbulent mixing stands out as a primary contributor to the model differences in inert tracers, especially in the relative contributions of BC1 and BC2 to

BCT. For the simulated mixing between PBL and the free troposphere, the differences in vertical grid structures, not only within the PBL but also in the free troposphere employed among models, can be as important as the differences in the representation of vertical turbulent mixing.

The results of the model inter-comparison in inert tracers also suggest that it should be essential to understand that when similar estimates on the impact of lateral boundary ozone are found between different simulations, the results do not

necessarily mean that the agreement has been reached for the same reason, unless a careful comparison is performed at the process level to rule out the possibility of cancelling process contributions. To carry out such analysis, process analysis (PA) (Jeffries and Tonnesen, 1994) is desired for all simulations involved. Unfortunately, the PA tool is either not available, or it was not turned on during the simulation, as PA is not a standard design for the AQMEII3 participants. We recommend the future model inter-comparison studies to include the PA tool as a standard design. The relatively higher contributions of the

BC2 tracer to BCT also suggest further examination of model concentrations and processes in the free troposphere and how those differences across models influence the surface predictions. Unfortunately, only minimal aloft data is available from the models participating in AQMEII3, precluding such an analysis here. Future model inter-comparison studies should



consider more detailed archiving of 3D model information, to enable a more complete diagnostic assessment of role of differing 3D transport representation on differences in surface level predictions across these modeling systems.

Though the four simulations have shown significant differences in the physical processes that the inert tracers undergo due to the model configurations and representations of physical processes, it could be speculated that if chemical degradation of $O_3$

imported into the domain was also considered, the differences in physical processes would become less important.

We think that whether the differences in physical processes would still be important for the impact of LBC ozone compared with chemical loss is case dependent. For example, it is found that the differences in simulated DM8A $O_3$ and DM8A BCT between WRF/CMAQ and WRF/CAMx (Figure 10) show strong spatial correlations and similar magnitudes (especially during winter and spring), suggesting that differences in representation of the 3D transport of LBC $O_3$ influence the noted

differences in surface $O_3$ predictions between the two models on a seasonal average basis.  During summer however the agreement is weaker and this is likely associated with the lack of representation of LBC $O_3$ chemical decay due to photolysis. During winter, spring and fall, the modeled differences in BCT are dominated by the difference in BC2. However, during summer, lateral boundary ozone from the mid-troposphere could decrease as the photolysis of ozone is most active in summer. In addition, the chemical formation of ozone also peaks in summer. Therefore, different from the other seasons, the

impact of lateral boundary ozone in summer due to physical processes become less important than chemical processes, so that the model difference in $O_3$ is not consistent with the difference in BCT anymore. Similar findings have also been reported by Baker et al. (2015), in which the chemically reactive and inert tracers were compared by season.

LBC $O_3$ could also be lost due to titration by NOx especially within the boundary layer. The contrast in DM8A $O_3$ between rural and urban sites is compared amongst models to indicate the model similarity/difference in chemical environment at

surface. WRF/CMAQ and WRF/CAMx is found to have very similar contrast between rural and urban sites (Figure 11), suggesting that the impact of lateral boundary ozone depletion at surface could be similar between the two models as well.

**Acknowledgements and Disclaimer**

We gratefully acknowledge the Air Quality Model Evaluation International Initiative (AQMEII) for facilitating the analysis described in the manuscript by designing and coordinating internally consistent regional-scale air quality model simulations.

During the conduct of this work, P. Liu held National Research Council post-doctoral fellowships. The views expressed in this article are those of the authors and do not necessarily represent the views or policies of the U.S. Environmental Protection Agency. Aarhus University gratefully acknowledges the 604 NordicWelfAir project funded by the NordForsk's Nordic Programme on Health and Welfare 605 (grant agreement no. 75007), the REEEM project funded by the H2020-LCE Research and 606 Innovation Action (grant agreement no.: 691739), and the Danish Centre for Environment 607 and Energy

(AU-DCE).    The simulation of WRF/CAMx was supported by the Coordinating Research Council Atmospheric Impacts Committee.



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



**List of Tables**

Table 1. Model description of the four simulations involved in the study





**List of Figures**

Figure 1. The studied area for this research, and the observation sites for ozone available in the data system of ENSEMBLE, with black dots for urban sites and red dots for rural sites.

Figure 2. Vertical grid resolution for chemical transport models used in the four modeling systems.

Figure 3. Seasonal average of the DM8A BCT from WRF/CMAQ (first row), and the difference in DM8A BCT between WRF/CAMx and WRF/CMAQ, between WRF/DEHM and WRF/CMAQ, and between COSMO-CLM/CMAQ and WRF/CMAQ (from the second to the fourth row) at all receptors. Results shown are as other model minus WRF/CMAQ.

The last row is the difference in DM8A BCT between WRF/DEHM_NODDEP and WRF/CMAQ_NODDEP (WRF/DEHM_NODDEP minus WRF/CMAQ_NODDEP).

Figure 4. Seasonal averages from WRF/CMAQ of (DM8A BC1)/(DM8A BCT) (first row), DM8A BC1 (second row), and DM8A BC2 (third row) at all receptors. The last three rows are the same as the first three rows but for

WRF/CMAQ_NODDEP.

Figure 5. Seasonal average of the dry deposition velocity of ozone from WRF/CMAQ (first row), and the difference between WRF/CAMx and WRF/CMAQ (WRF/CAMx minus WRF/CMAQ) (second row).

Figure 6. Difference between WRF/CAMx and WRF/CMAQ (WRF/CAMx minus WRF/CMAQ) in the seasonal averages of (DM8A BC1)/(DM8A BCT) (first row), DM8A BC1 (second row), and DM8A BC2 (third row) at all receptors.

Figure 7. Difference between WRF/DEHM and WRF/CMAQ (WRF/DEHM minus WRF/CMAQ) in the seasonal averages of (DM8A BC1)/(DM8A BCT) (first row), DM8A BC1 (second row), and DM8A BC2 (third row) at all receptors. The last

three rows are the same as first three rows but for the difference between WRF/DEHM_NODDEP and WRF/CMAQ_NODDEP (WRF/DEHM_NODDEP minus WRF/CMAQ_NODDEP).

Figure 8. Seasonal average of the difference in PBL between WRF/DEHM and WRF/CMAQ at all receptors averaged by hourly data during daytime (first row). The difference in PBL is (WRF/DEHM minus WRF/CMAQ) and normalized by

WRF/CMAQ. The second row is the same as first row but between COSMO-CLM/CMAQ and WRF/CMAQ.



Figure 9. Difference between COSMO-CLM/CMAQ and WRF/CMAQ (COSMO-CLM/CMAQ minus WRF/CMAQ) in the seasonal averages of (DM8A BC1)/(DM8A BCT) (first row), DM8A BC1 (second row), and DM8A BC2 (third row) at all receptors.

5   Figure 10. Difference between WRF/CAMx and WRF/CMAQ (WRF/CAMx minus WRF/CMAQ) in the seasonal averaged DM8A BCT, and DM8A ozone at all receptors.

Figure 11. Difference in DM8A ozone between rural and urban sites (rural minus urban) from observation, WRF/CMAQ, and WRF/CAMx.



**Table 1: Model description of the four simulations involved in the study**

|  | WRF/CMAQ | WRF/CAMx | COSMO-CLM/CMAQ | WRF/DEHM |
|---|---|---|---|---|
| Institute | U.S. EPA | RAMBOLL Environ (U.S.) | Helmholtz-Zentrum Geesthacht (Germany) | Aarhus University (Denmark) |
| Global Meteorology | NCEP | NCEP | NCEP | ECMWF |
| Regional Modeling System | WRF3.4/ CMAQ5.0.2 | WRF3.4/ CAMx6.2 | COSMO-CLM/ CMAQ5.0.1 | WRF/ DEHM |
| Horizontal Resolution | 12km | 12km | 24km | 50km |
| Gas Phase Chemistry | CB05-TUCL | CB05 | CB05-TUCL | Brandt et al. (2012) |
| Dry Deposition for Ozone and Tracers | Pleim and Ran (2011) | Zhang et al. (2003) | Pleim and Ran (2011) | Simpson et al. (2003) |
| Wet Deposition for Ozone | YES | YES | YES | NO |
| Wet Deposition for Tracers | YES | NO | YES | YES |
| Impact of Sub-Grid Clouds on Radiation in RCMs | NO | NO | YES | NO |
| Impact of Sub-Grid Clouds on Ozone Photochemistry in CTMs | YES | YES | YES | YES |
| Sub-Grid Cloud Mixing in CTMs | YES | NO | YES | YES |



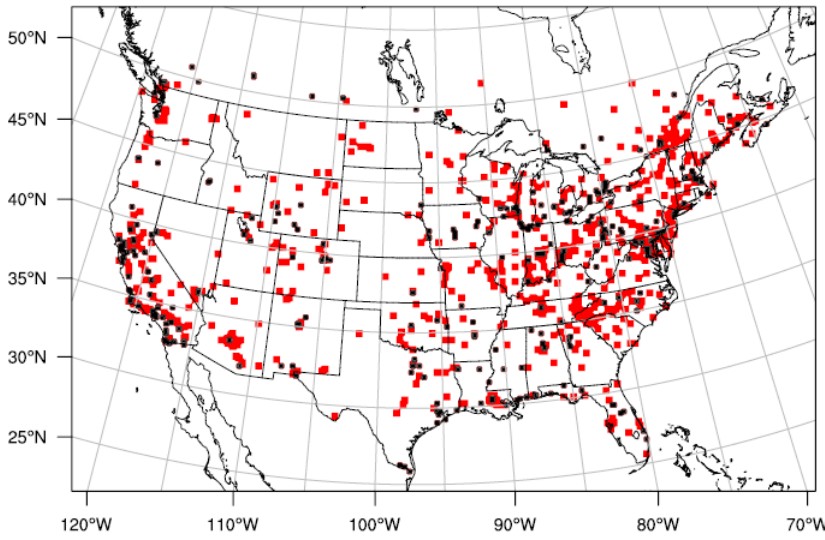

**Figure 1: The studied area for this research, and the observation sites for ozone available in the data system of ENSEMBLE, with black dots for urban sites and red dots for rural sites.**

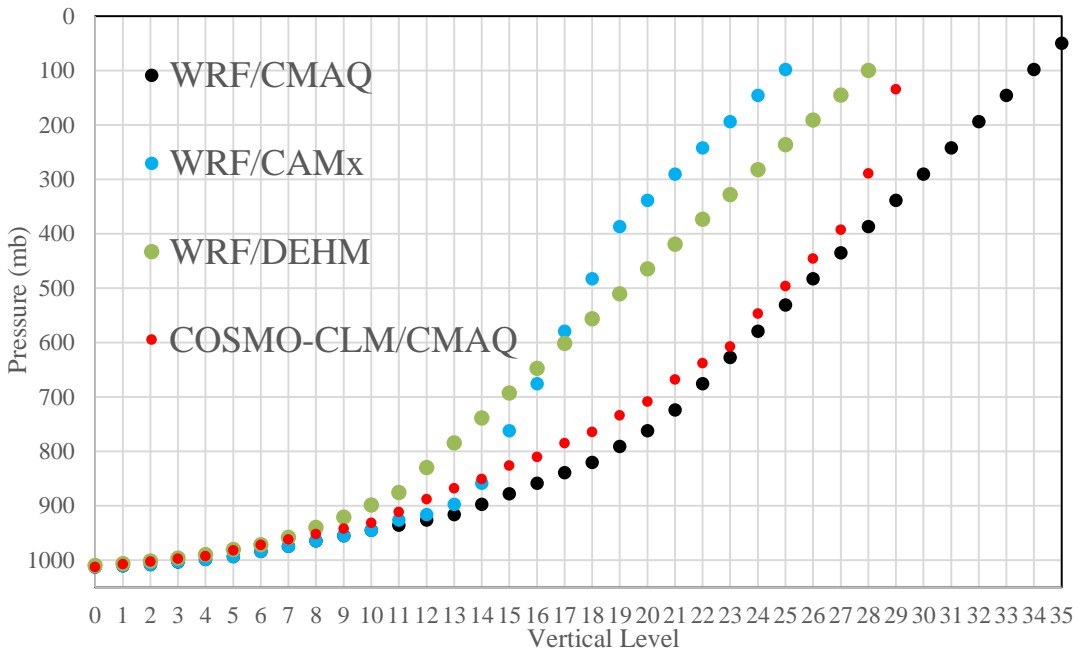

**Figure 2: Vertical grid resolution for chemical transport models used in the four modeling systems.**





**Figure 3: Seasonal average of the DM8A BCT from WRF/CMAQ (first row), and the difference in DM8A BCT between WRF/CAMx and WRF/CMAQ, between WRF/DEHM and WRF/CMAQ, and between COSMO-CLM/CMAQ and WRF/CMAQ (from the second to the fourth row) at all receptors. Results shown are as other model minus WRF/CMAQ. The last row is the difference in DM8A BCT between WRF/DEHM_NODDEP and WRF/CMAQ_NODDEP (WRF/DEHM_NODDEP minus WRF/CMAQ_NODDEP).**





**Figure 4: Seasonal averages from WRF/CMAQ of (DM8A BC1)/(DM8A BCT) (first row), DM8A BC1 (second row), and DM8A BC2 (third row) at all receptors. The last three rows are the same as the first three rows but for WRF/CMAQ_NODDEP.**





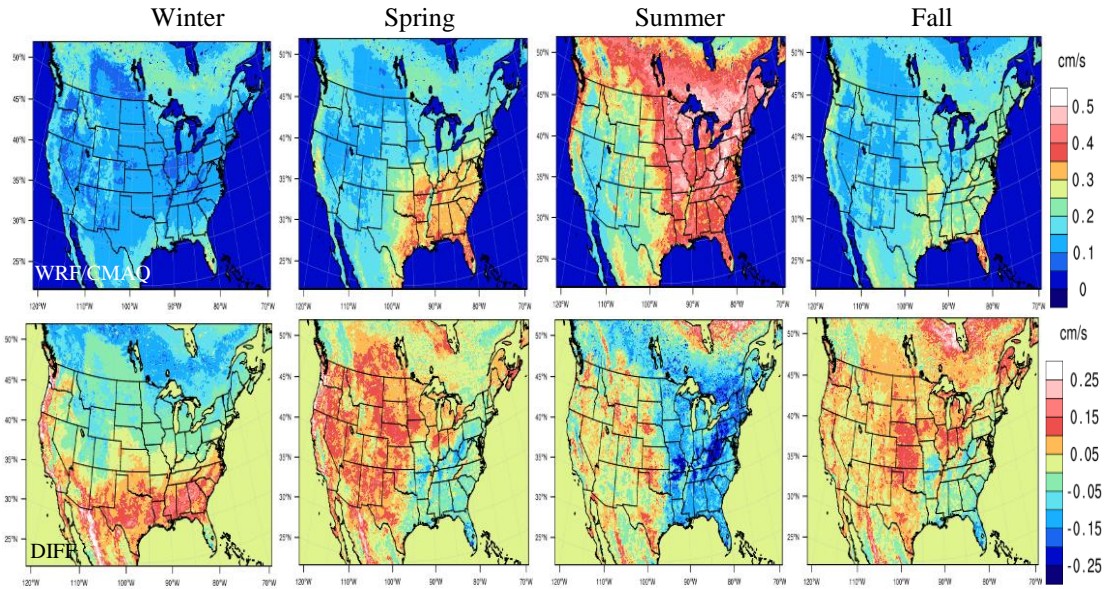

**Figure 5: Seasonal average of the dry deposition velocity of ozone from WRF/CMAQ (first row), and the difference between WRF/CAMx and WRF/CMAQ (WRF/CAMx minus WRF/CMAQ) (second row).**





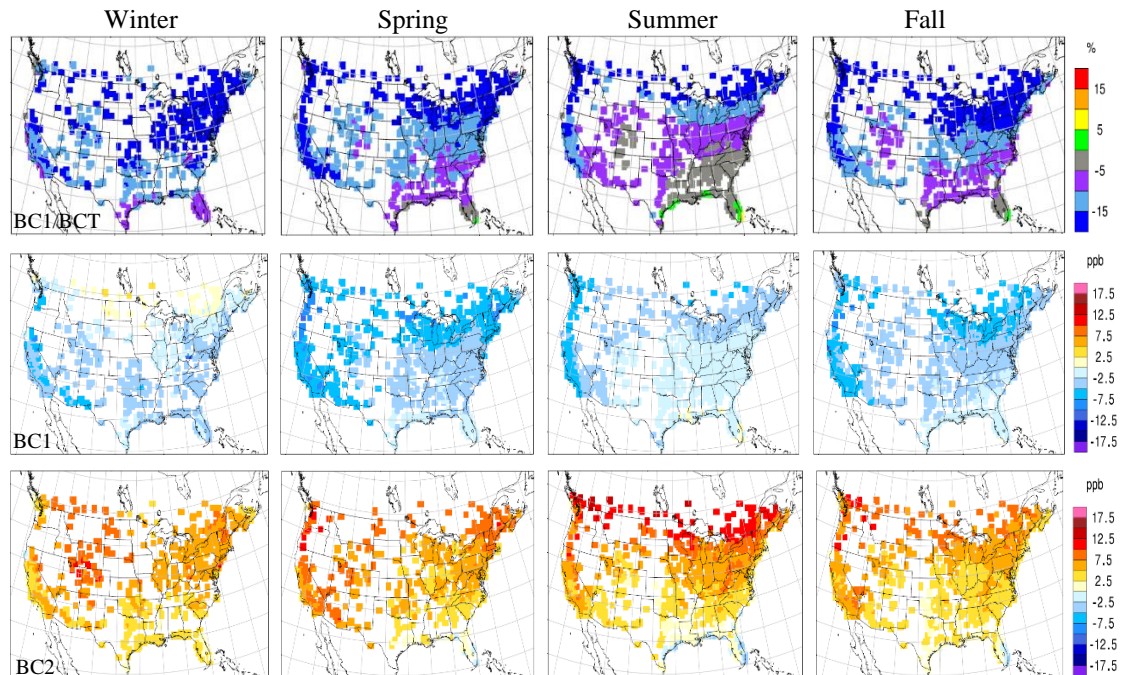

**Figure 6: Difference between WRF/CAMx and WRF/CMAQ (WRF/CAMx minus WRF/CMAQ) in the seasonal averages of (DM8A BC1)/(DM8A BCT) (first row), DM8A BC1 (second row), and DM8A BC2 (third row) at all receptors.**



**Figure 7: Difference between WRF/DEHM and WRF/CMAQ (WRF/DEHM minus WRF/CMAQ) in the seasonal averages of**
**(DM8A BC1)/(DM8A BCT) (first row), DM8A BC1 (second row), and DM8A BC2 (third row) at all receptors. The last three rows**
**are the same as first three rows but for the difference between WRF/DEHM_NODDEP and WRF/CMAQ_NODDEP**
**(WRF/DEHM_NODDEP minus WRF/CMAQ_NODDEP).**





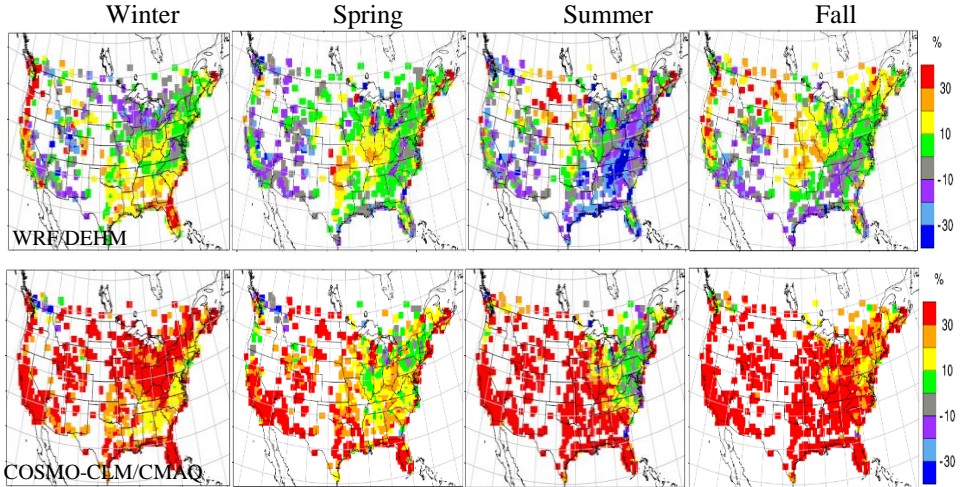

**Figure 8: Seasonal average of the difference in PBL between WRF/DEHM and WRF/CMAQ at all receptors averaged by hourly data during daytime (first row). The difference in PBL is (WRF/DEHM minus WRF/CMAQ) and normalized by WRF/CMAQ. The second row is the same as first row but between COSMO-CLM/CMAQ and WRF/CMAQ.**

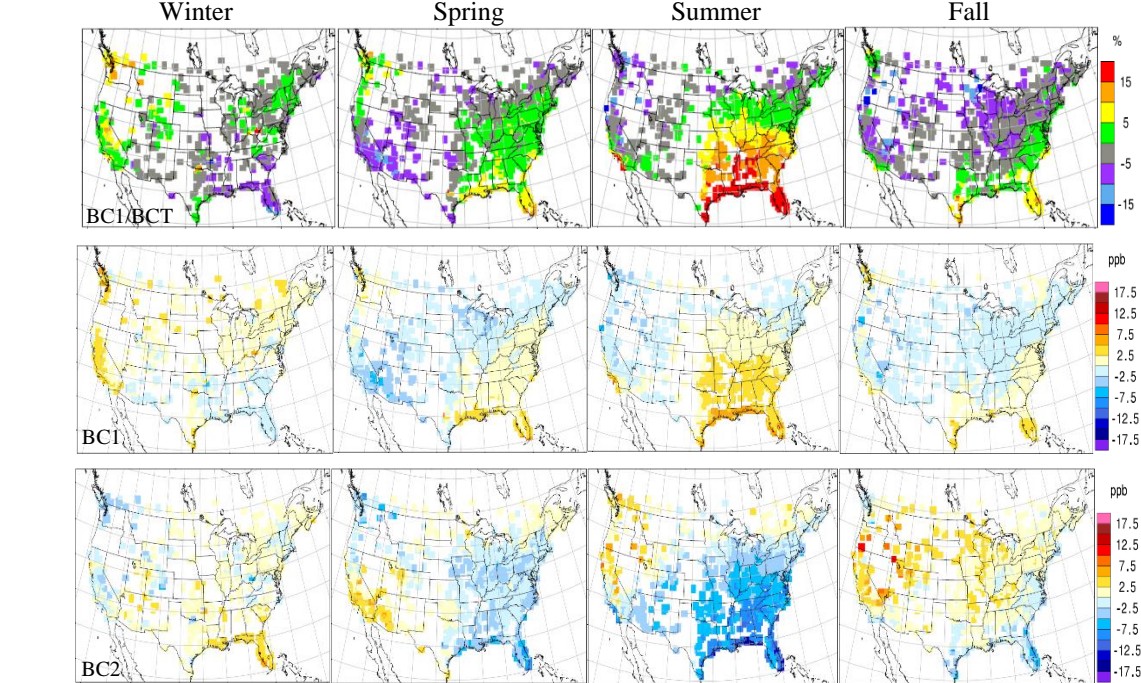

**Figure 9: Difference between COSMO-CLM/CMAQ and WRF/CMAQ (COSMO-CLM/CMAQ minus WRF/CMAQ) in the seasonal averages of (DM8A BC1)/(DM8A BCT) (first row), DM8A BC1 (second row), and DM8A BC2 (third row) at all receptors.**



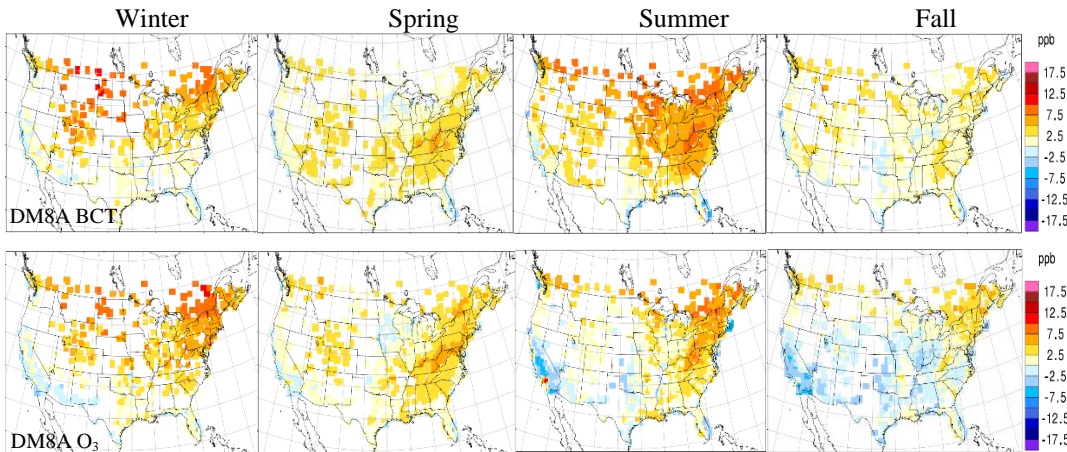

**Figure 10: Difference between WRF/CAMx and WRF/CMAQ (WRF/CAMx minus WRF/CMAQ) in the seasonal averaged DM8A BCT, and DM8A ozone at all receptors.**

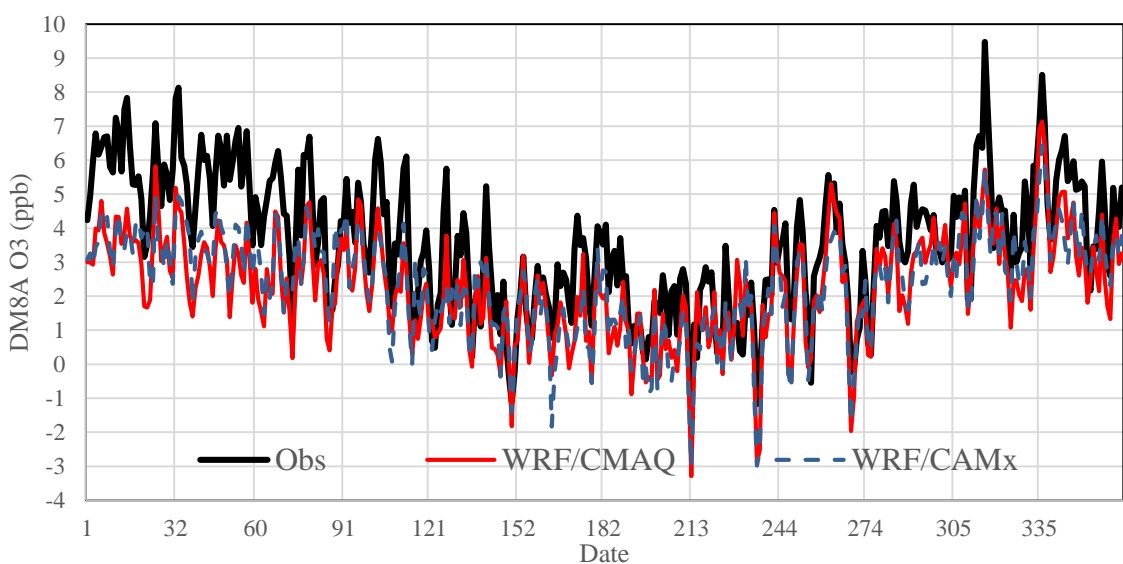

10 **Figure 11: Difference in DM8A ozone between rural and urban sites (rural minus urban) from observation, WRF/CMAQ, and WRF/CAMx.**