# Peer review of "Attributing Differences in the Fate of Lateral Boundary Ozone in AQMEII3 Models to Physical Process Representations"

_Atmospheric Chemistry and Physics, 2018_

## Referee Comment (RC1) · Anonymous Referee #1 · 24 Apr 2018

In "Multi-Model Comparison in the Impact of Lateral Boundary Conditions on Simulated Surface Ozone across the United States Using Chemically Inert Tracers", Liu et al. described the results of lateral-boundary chemically inert tracer implemented in four CTMs with different configurations under the model-intercomparison project AQMEII3, to understand the inter-model differences in the impact of lateral boundary conditions on simulated surface ozone. The method presented in the manuscript is novel and interesting; however, I think the manuscript still needs substantial improvements (especially in writing and presentation) before publication in ACP.

Major comments: The "Results" (Section 3) are not well presented. This section reads

more like a report rather than a research paper. A read may have a hard time finding the major messages (as well stated in the abstract) that the authors wanted to convey, because 1) The content is organized with model pairs, but without clearly stating at the beginning of each subsection what we can learn from the two different setups, which makes the following discussion confusing. 2) The discussions are not focused on the major conclusions (and maybe too lengthy for subsections). Many descriptions and discussions of the minor temporal and spatial features, in my opinion, should be removed or reorganized, especially those cannot be readily explained by current model pairs. 3) The multi-panel color-coded maps are difficult for a reader to follow (especially there are 7 of them) and sometimes misleading (because color scales are different). Since the paper mainly focuses on the difference between mountain west and eastern US, the authors may consider to summarize the results in two regions and present it to the main points of their discussion. 4) Discussions are mostly descriptive and qualitative. Readers may be interested in quantitative estimations of the uncertainties associated with model configurations and physical processes.

I think the authors can greatly improve the paper by rethinking the organization of information and discussion in this section.

Minor comments: Page 7 Line 18-20; Line 31-32: I am not convinced that the higher BC2 (lower BC1) in the mountain west is due to vertical mixing between mid-troposphere and the PBL. More likely, the existence of high mountains can divert the wind in BC1-enriched lower troposphere, so there is less BC1 advected to mountain regions, which is very different from vertical mixing.

Page 7 Line 21: "One possible explanation is that the impact from vertical mixing and the impact from LBCs cancel out in the mountain west region.Âă" Are all these coming from lateral boundary conditions as the authors are analyzing lateral boundary inert tracer? What exactly is the "impact from LBC" here referring to?

Table 1: Do the differences in "gas phase chemistry", "wet deposition for tracers", "impact of sub-grid clouds on radiation", and "impact of sub-grid clouds on ozone photo-chemistry" affect BC1 and BC2 results? What are the vertical turbulent mixing schemes for different model, which are discussed in the paper?

Figure 10: The authors may consider to move figure 10 to main results. It shows nicely why this study is important.

---

## Referee Comment (RC2) · Anonymous Referee #2 · 10 May 2018

Overview:

The paper presents a multi-model comparison of simulated surface values over North America of chemically inert tracers released at different height levels at the boundaries of the region. The tracers represent the boundary conditions for ozone in the PBL (BC1), the free troposphere (BC2) and the upper troposphere/lower stratosphere (BC3). The tracers are subject to dry and wet deposition assuming the specification of ozone. The authors find differences in the BC tracer concentrations at the surface and there relative contribution between the four models. These differences are mainly attributed to differences in the simulation of vertical turbulent mixing by the four models.

[Figure]

Interactive
comment

General and specific remarks

The title of the paper refers to an interesting scientific question, i.e. the impact of BC for surface ozone. However, the paper does not really provide useful answers to that question. My main concerns are as follow, which are further elaborated below:

- The simulated tracers, which are meant to indicate the impact of BC ozone on surface ozone, are not compared in any way to the simulated surface ozone values of the models.

- The tracer results are not juxtaposed to observations (or model errors) in any way despite the surprising fact that the tracer results are shown only for the location of the ozone observations.

- A BC tracer which is meant to mimic ozone should be subject to an equivalent chemical loss, which is not the case in this study.

- The impact of the top boundary is not discussed despite that fact that stratospheric ozone can influence surface ozone in certain situations.

- The choice of the DMA8 average needs to be better motivated as it excludes night time values, when differences in mixing can be large between different models.

- The difference between the models are discussed in a lot of detail, which is overwhelming for the reader

The title of the study suggest that the paper will give an estimate of the impact of the BC on simulated surface ozone. I did not find any such estimate in the paper. The simulated ozone surface concentrations are not compared to the BC tracer results in anyway. I strongly suggest to change the title of the paper to avoid giving a false impression of its content.

Although artificial tracers cannot be directly evaluated with observations, an analysis of the model errors for ozone in relation to the tracer simulation should have been an

[Figure]

**[ACPD](ACPD)**

Interactive
comment

interesting aspect of the paper. On the other hand, the paper does give the impression that observations are considered because the observations are discussed (figure 1, section 2.3) and all plots of tracer simulation (figures 3-4, 6-10) are provided at the sampling location of the observations. But there is no use of this, which is confusing. Producing continuous maps of the tracers simulations would have been more appropriate. The only figure (11) showing a comparisons with observation is a time series of mean differences between rural and urban observations. This could be an interesting figure but it is just mentioned in passing with no detailed discussion.

Getting a better knowledge of the impact of boundary ozone concentrations, both from a modelling or more factual point of view, on surface ozone over North-America is an interesting question. Using dedicated tracer simulations is an appropriate method to investigate the problem. The authors apply tracers that are only subject to dry and wet deposition but not the chemical ozone loss. It is debatable – as also pointed out by the authors in the conclusion - if this approach does full justice to the problem of ozone because chemical ozone loss could be an important factor for the importance of BC on surface ozone. A quantitative comparison of the contribution of (i) chemical loss, (ii) loss by dry deposition and (iii) by wet deposition to the ozone budget in the study area for the different models would have been needed to demonstrate if the used approach is reasonable or not. (It would probably also show that wet deposition of ozone is of minor importance). Truly inert BC tracers, i.e. also without deposition, give some indication of the transport processes only. They would be not specific to the characteristics of ozone (apart from the BC values) but could still give some information. Hence, a better discussion of the runs without dry deposition could be helpful.

The authors find very little transport from the lateral BC to the surface for tracer BC3 (< 250 hPa). It needs to be discussed in more detail, if this means that stratospheric ozone does not impact surface ozone in the study area at all. However, the top boundary condition would need to be considered for such an investigation. Considering only lateral influx with BC3, given that the models had only 1, 3 or 5 model levels above 250

hPa, seems not sufficient. A realistic top boundary is required to simulate the impact of upper tropospheric and stratospheric ozone on surface ozone.

A clarification of the setup of the BC tracers would be helpful. Did the values of the ozone BC from the global model agree with observations ? Where the BC placed at exactly the same position for all models or did the location of the model boundaries vary from model to model? This could have had an impact on the results.

The main finding of the authors is that vertical mixing differs between the models. It would be interesting to elaborate on the reasons for this. The authors could provide a confirmation of this finding by studying profiles of primary species emitted over the domain. Also, a statement on which of the different mixing scheme leads to the best results would give useful information to the reader.

DMA8 is the only time aggregation of the discussed BC results. The choice of DMA8 for the given applications needs to be better motivated. As the focus of the study is to identify differences in vertical mixing, night-time values should not be disregarded as it is most likely the case for DMA8 of ozone. It is during night time that different vertical mixing (or the lack thereof) can have the largest impact on surface values. Likewise it would be interesting to check if the time period for the calculation of DM8A was the same for the models. Differences in the diurnal cycle can be an indication of different simulation of vertical mixing.

Most of the paper is dedicated to a detailed comparison between different model versions. This is very detailed and can be tedious for a reader, who is not a developer of one of the discussed models. Given this highly technical aspect of the paper, I recommend publication in another, more suited journal such as GMD.

The colour scale for absolute values could be improved. In my printout the magenta (e.g. 25-30 ppb range, top, Fig 3) looks very similar to the red colours indicating high values.

---

## Author Comment (AC1) · 30 Oct 2018

Response to referee #1

We would like to thank the reviewer for providing constructive comments that have helped us improve the manuscript. The reviewer comments are shown in regular font, our responses are shown in bold fond, and a summary of the changes made to the manuscript in response to each comment is shown in italics.

Major comments

The content in results section is organized with model pairs, but without clearly starting at the beginning of each subsection with what we can learn from the two different setups, which makes the following discussion confusing.

**Response: We agree with the reviewer that adding a statement about what can be learned from a given model pair at the beginning at each subsection would make the paper easier to follow.**

*Changes in manuscript: Such statements have been added at the beginning of the comparison between different model pairs. This study aims at understanding the model variability originating from the physical processes in CTMs and its impact on the inert tracers of lateral boundary ozone reaching the surface. The first model pair (WRF/CMAQ versus WRF/CAMx) used the same meteorological inputs for CTMs (and therefore the same representation of advection) but different representations for the other important physical processes that the inert tracers undergo (e.g. vertical mixing and dry deposition). The situation is the opposite for the second model pair (WRF/CMAQ versus COSMO-CLM/CMAQ). Therefore, the results from the two model pairs serve as a good example for demonstrating the relative importance of 3D advection and the other physical processes in CTMs on the model variability in inert tracers at surface.*

The discussions in results section are not focused on the major conclusions (and maybe too lengthy for subsections). Many descriptions and discussions of the minor temporal and spatial features. In my opinion, should be removed or reorganized, especially those cannot be readily explained by current model pairs.

**Response: We agree that many descriptions and discussions were lengthy and not sufficiently focused on the key messages. Moreover, due to lack of extra sensitivity simulations, some discussions were not supported firmly by the simulations presented in the original manuscript. Therefore, three sensitivity simulations have been conducted for WRF/CMAQ to investigate the impact of different physical processes on the inert tracers at the surface. These physical processes are important processes that the inert tracers undergo in CTMs, including wet and dry deposition and sub-grid cloud mixing. In addition, one more sensitivity simulation has been carried out to investigate the impact of vertical grid structure on inert tracers at the surface. With the help of these sensitivity simulations for WRF/CMAQ, the model differences in inert tracers between WRF/CMAQ and the other models can be tied to specific processes in the revised manuscript. Ideally, the sensitivity simulations conducted for WRF/CMAQ would also have been performed for the other three models. Unfortunately, however, since these sensitivity simulations were not part of the AQMEII3 protocol most sensitivity simulations are not available for the other three models.**

*Changes in manuscript: The results from these sensitivity simulations have been included in the revised paper. The results have also been reorganized so that the results are discussed with respect to the impact of each physical process on the total inert tracers at the surface and on the relative contributions of inert tracers from different altitude ranges, respectively.*

The multi-panel color-coded maps are difficult for a reader to follow (especially there are 7 of them) and sometimes misleading (because the color scales are different). Since the paper mainly focuses on the difference between mountain west and eastern US, the authors may consider to summarize the results in two regions and present it to the main points of their discussion.

**Response: We agree that the panel plots showing the entire domain may have made the discussions hard to follow. As suggested by the reviewer, results are shown for sub-regions in the revised manuscript.**

*Changes in manuscript: In the revised paper, seven sub-regions were selected from the domain based on their proximity to the lateral boundaries, elevations, and climate, including WB (region close to western boundary), NB (region close to northern boundary), MT (west mountain area), GP (great plain area), NE (northeast), SE (southeast), and ATL (the Atlantic Ocean). When explaining the impact of physical processes on the inert tracers at surface, the results (including the vertical profiles and diurnal cycles of inert tracers) are averaged over sub-regions, so that readers can clearly see the different/similar roles that certain physical process plays in a variety of regions over the U.S. Some of the panel plots of the entire domain are still kept and reorganized to show the general differences between different simulations and two tables have been added to summarize the averaged results over land. The color scales are kept the same when showing the same metric in different sub-sections in the results.*

Discussions are mostly descriptive and qualitative. Readers may be interested in quantitative estimates on the uncertainty associated with the model configurations and physical processes.

**Response: Due to the limited data available to us at the time the original manuscript was prepared, it had not been possible for us to provide more quantitative estimates regarding the uncertainty associated with the model configurations and physical processes. However, we agree with the reviewer that providing more quantitative estimates would be of interest to the reader. Therefore, we have carried out extra sensitivity simulations for WRF/CMAQ to quantify the effects of different process representations on the fate of inert boundary condition tracers. Ideally, the sensitivity simulations conducted for WRF/CMAQ would also have been performed for the other three models. Unfortunately, however, since these sensitivity simulations were not part of the AQMEII3 protocol, , most sensitivity simulations are not available for the other three models.**

*Changes in manuscript: In the revised paper, the uncertainty of inert tracers at the surface associated with the model configurations and physical processes has been quantitatively estimated for WRF/CMAQ. In addition, with the help of the extra sensitivity simulations for WRF/CMAQ, more quantitative estimates are also provided explaining the roles of model configurations and physical processes in the differences in inert tracers between WRF/CMAQ and the other models.*

Minor comments

Page 7 Line 18-20; Line 31-32: I am not convinced that the higher BC2 (lower BC1) in the mountain west is due to vertical mixing between mid-troposphere and the PBL. More likely, the existence of high mountains can divert the wind in BC1-enriched lower troposphere, so there is less BC1 advected to mountain regions, which is very different from vertical mixing.

**Response: We agree that this is a plausible alternative explanation for this behavior.**

*Changes in manuscript: This statement has been removed in the revised paper. With the current data, no further analysis can be made to determine whether the higher BC2 and lower BC1 over the mountain is due to vertical mixing or the diversion of the wind. In addition, in the revised paper, we removed the discussions built on the results of WRF/CMAQ alone. Instead, we focused on the comparison between WRF/CMAQ and the extra sensitivity simulations for WRF/CMAQ to quantitatively estimate the uncertainty related to model configurations and physical processes, so that our points are much better supported by the results and presented to the readers more clearly.*

Page 7 Line 21: "One possible explanation is that the impact from vertical mixing and the impact from LBCs cancel out in the mountain west region." Are all these coming from lateral boundary conditions as the authors are analyzing lateral boundary inert tracer? What exactly is the "impact from LBC" here referring to?

**Response: Yes, they all come from lateral boundary as we are analyzing the lateral boundary inert tracers reaching the surface of the U.S. The levels of inert tracers at the surface are affected by 3D advection, vertical turbulent mixing and some other physical processes. Here, the "impact from LBC" refers to the inert tracers coming to the mountain area through horizontal advection. As explained in the following sentence, since the western boundary is predominantly an inflow boundary for lateral boundary inert tracers, it should be expected that BC1 in the west is higher than that in the east due to horizontal advection. We agree that this statement is confusing and should be modified.**

*Changes in manuscript: In the revised paper, we removed such discussions built on the results of WRF/CMAQ alone. Instead, we focused on the comparison between WRF/CMAQ and the extra sensitivity simulations for WRF/CMAQ to quantitatively estimate the uncertainty related to model configurations and physical processes, so that our points are much better supported by the results and presented to the readers more clearly.*

Table 1: Do the differences in "gas phase chemistry", "wet deposition for tracers", "impact of sub-grid clouds on radiation", and "impact of sub-grid clouds on ozone photochemistry" affect BC1 and BC2 results? What are the vertical turbulent mixing schemes for different model, which are discussed in the paper?

**Response: The differences in "gas phase chemistry", "impact of sub-grid clouds on radiation", and "impact of sub-grid clouds on ozone photochemistry" do not affect BC1 and BC2 since they are chemically inert tracers. The differences in "wet deposition for tracers" might affect BC1 and BC2 since it is one of the physical processes that the inert tracers would undergo. The vertical turbulent mixing schemes used in the CTMs are ACM2 for WRF/CMAQ and COSMO-CLM/CMAQ, and K-theory for WRF/CAMx WRF/DEHM.**

*Changes in manuscript: Since our paper focuses on the physical treatment in CTMs and its impact on inert tracers of lateral boundary ozone, the content in Table 1 has been revised to only include the physical processes that the inert tracers undergo, including dry and wet deposition, sub-grid cloud mixing and vertical turbulent mixing in CTMs. The processes related to ozone chemistry have been removed.*

Figure 10: The authors may consider to move figure 10 to main results. It shows nicely why this study is important.

**Response: In the original paper, Figure 10 was placed in "summary and discussion" to demonstrate the relative importance of physical processes versus the chemical processes on the lateral boundary ozone. As the reviewer suggested, we have moved it to main results to show the important impacts of physical processes on the lateral boundary ozone reaching the surface.**

*Changes in manuscript: This Figure has been moved to the section of results, under the sub-section of WRF/CMAQ versus WRF/CAMx.*

Response to referee #2

We would like to thank the reviewer for providing constructive comments that have helped us improve the manuscript. The reviewer comments are shown in regular font, our responses are shown in bold fond, and a summary of the changes made to the manuscript in response to each comment is shown in italics.

The simulated tracers, which are meant to indicate the impact of BC ozone on surface ozone, are not compared in any way to the simulated surface ozone values of the models.

**Response: No, the results of inert tracers for lateral boundary ozone are not compared with the simulated ozone values, because quantifying model performance for ozone is beyond the scope of this study. As the reviewer subsequently pointed out, the title of the original manuscript may have been misleading, leading readers to expect a quantitative estimate of the contributions of lateral boundary ozone to ozone levels at the surface. However, this is not the goal of our paper.**

*Changes in manuscript: To avoid the confusion, the title has been changed to "Attributing Differences in the Fate of Lateral Boundary Ozone in AQMEII3 Models to Physical Process Representations". In addition, the introduction has been modified to emphasize that this paper focuses on quantifying the model variability originating from the physical processes in CTMs, and aims at understanding how different representation of physical processes in CTMs may lead to the differences in the LB ozone that eventually reaches the surface across the U.S.*

The tracer results are not juxtaposed to observations (or model errors) in any way despite the surprising fact that the tracer results are shown only for the location of the ozone observations.

**Response: We agree with the reviewer that since the modeled results had not been compared with observations in the original manuscript, it did not make sense that the results were shown only at the locations where ozone observations are available. The reason for displaying the model data in this fashion in the original manuscript was that such data extracted at monitoring locations was readily available while gridded model fields were not. However, such gridded model fields have now been obtained from all participating groups and are used in the revised manuscript.**

*Changes in manuscript: Instead of using the AQMEII3 dataset at ozone receptors, gridded model fields have been obtained and are used in the revised paper. To generate this gridded data set, participating groups re-gridded the modeled hourly values for inert tracers at the surface to a common domain for analysis and comparison, covering the area from 23.5⁰ N/-130.0⁰ W to 58.5⁰ N/ -59.5⁰ W (green shaded area in Figure 1a in the revised paper) with grid spacing of 0.25⁰ by 0.25⁰. Spatially continuous maps of the model differences in inert tracers at surface are shown in the revised paper.*

A BC tracer which is meant to mimic ozone should be subject to an equivalent chemical loss, which is not the case in this study.

**Response: The reviewer is correct that a BC tracer aimed at quantitatively attributing surface ozone to BC should include the chemical loss of ozone. Several studies have used reactive tracers to estimate the contributions of lateral boundary ozone to the ozone level at surface over the U.S. However, few studies have investigated the uncertainty in the impact of lateral boundary ozone on the ozone levels at the surface. The goal of this paper is to focus on the model variability originating from the physical processes in CTMs, and it aims at understanding how different representation of physical processes in**

**CTMs may lead to the differences in the LB ozone that eventually reaches the surface across the U.S. This research goal can be fulfilled by using chemically inert tracers for lateral boundary ozone.**

*Changes in manuscript: Modifications have been made in the introduction section to emphasize the research goal of this study, and we also clarified that "it is necessary to include the chemical loss of LB ozone when quantitatively estimating the impact of LB ozone, as shown in the comparison between inert and reactive LB ozone tracers by Baker et al. (2015)" to avoid any misunderstanding in the use of inert tracers.*

The impact of the top boundary is not discussed despite that fact that stratospheric ozone can influence surface ozone in certain situations.

**Response: We agree that stratospheric ozone can influence surface ozone in certain situations. However, quantifying the contributions of stratospheric ozone to surface ozone across the U.S. is outside the scope of this paper. In terms of the influence of the top boundary on simulation results, we also refer the reviewer to our response to comment #11.**

*Changes in manuscript: Modifications have been made in the title and in the introduction section to emphasize the research goal of this study, so that the readers do not expect a quantitative estimate of the contributions of ozone from lateral boundaries or stratospheric intrusion to the ozone levels at the surface.*

The choice of the DMA8 average needs to be better motivated as it excludes night time values, when differences in mixing can be large between different models.

**Response: We agree that it may be also important to look at the model differences during the night time.**

*Changes in manuscript: In the revised paper, the diurnal cycles of the model differences in the inert tracers at surface are also investigated, and the results are shown by averaging over sub-regions on a seasonal basis.*

The difference between the models are discussed in a lot of detail, which is overwhelming for the reader.

**Response: We agree that the original manuscript included a lot of details that sometimes may have distracted the reader from the main points.**

*Changes in manuscript: Some minor features in the temporal and spatial differences in inert tracers between different models have been removed. Instead, in the revised paper, when comparing the model differences in inert tracers, the discussion is reorganized so that the impact of different physical treatment in CTMs on inert tracers are discussed individually, including vertical resolution, wet scavenging, dry deposition, sub-grid cloud mixing, and vertical turbulent mixing.*

The title of the study suggest that the paper will give an estimate of the impact of the BC on simulated surface ozone. I did not find any such estimate in the paper. The simulated ozone surface concentrations are not compared to the BC tracer results in anyway. I strongly suggest to change the title of the paper to avoid giving a false impression of its content.

**Response: We agree that the title of the original manuscript may have been misleading and may have led the reader to expect an estimate of the impact of the BC on simulated surface ozone, which is not the goal of this study.**

*Changes in manuscript: The title has been changed to 'Attributing Differences in the Fate of Lateral Boundary Ozone in AQMEII3 Models to Physical Process Representations".*

Although artificial tracers cannot be directly evaluated with observations, an analysis of the model errors for ozone in relation to the tracer simulation should have been an aspect of the paper.

**Response: We agree that an analysis of the model errors for ozone in relation to the tracer simulation could be interesting. However, such analysis would not provide information about the model variability associated with the model configurations and physical processes in the lateral boundary ozone reaching the surface, which is the goal of this research. Therefore, such analysis is not included in the paper. However, we agree that the differences in tracers can sometimes explain differences in ozone concentrations, highlighting the importance of this study. In particular, it was found that the differences in DM8A $O_3$ and DM8A BCT between WRF/CMAQ and WRF/CAMx show strong spatial correlations with similar magnitudes except for summer, as shown in Figure 10 in the original paper (now Figure 7 in the revised paper).**

*Changes in manuscript: The Figure 10 in the original paper has been moved to the main results section under sub-section WRF/CMAQ versus WRF/CAMx as part of Figure 7.*

On the other hand, the paper does give the impression that observations are considered because the observations are discussed (figure 1, section 2.3) and all plots of tracer simulation (figures 3-4, 6-10) are provided at the sampling location of the observations. But there is no use of this, which is confusing. Producing continuous maps of the tracers simulations would have been more appropriate. The only figure (11) showing a comparison with observation is a time series of mean differences between rural and urban observations. This could be an interesting figure but it is just mentioned in passing with no detailed discussion.

**Response: We agree with the reviewer. Please see our response to comment #2 regarding the use of gridded model fields in the revised manuscript.**

*Changes in manuscript: Instead of using the AQMEII3 dataset at ozone receptors, gridded model fields have been obtained and are used in the revised paper. To generate this gridded data set, participating groups re-gridded the modeled hourly values for inert tracers at the surface to a common domain for analysis and comparison, covering the area from 23.5⁰ N/-130.0⁰ W to 58.5⁰ N/ -59.5⁰ W (green shaded area in Figure 1a in the revised paper) with grid spacing of 0.25⁰ by 0.25⁰. Spatially continuous maps of the model differences in inert tracers at surface are shown in the revised paper.As to Figure 11 in the original paper, it has been removed in the revised paper as such discussion is not closely related to the main points of this paper.*

Getting a better knowledge of the impact of boundary ozone concentrations, both from a modelling or more factual point of view, on surface ozone over North-America is an interesting question. Using dedicated tracer simulations is an appropriate method to investigate the problem. The authors apply tracers that are only subject to dry and wet deposition but not the chemical ozone loss. It is debatable – as also pointed out by the authors in the conclusion - if this approach does full justice to the problem of ozone because chemical ozone loss could be an important factor for the importance of BC on surface ozone. A quantitative comparison of the contribution of (i) chemical loss, (ii) loss by dry deposition and (iii) by wet deposition to the ozone budget in the study area for the different models would have been needed to demonstrate if the used approach is reasonable or not. (It would probably also show that wet deposition of ozone is of minor importance). Truly inert BC tracers, i.e. also without deposition, give some indication of the transport processes only. They would be not specific to the characteristics of ozone (apart from the BC values) but could still give some information. Hence, a better discussion of the runs without dry deposition could be helpful.

**Response: We agree that to quantitatively estimate the contributions of lateral boundary ozone to the ozone levels at surface, the chemical loss of ozone must be included. This has also been confirmed by comparing the inert and reactive lateral boundary ozone tracers at the surface across the U.S. (Baker et al.,2015). However, the goal of this paper is not to provide such an estimate of the contributions of lateral boundary ozone. Instead, this paper focuses on quantifying the model variability originating from the physical processes in CTMs, and aims at understanding how different representation of physical processes in CTMs may lead to the differences in the LB ozone that eventually reaches the surface across the U.S. Using chemically inert tracers is an appropriate approach for accomplishing this goal. We agree with the reviewer that a quantitative comparison of the effects of different processes that the inert tracers undergo would be valuable. Therefore, extra sensitivity simulations for WRF/CMAQ have been carried out to provide quantitative estimates of the uncertainty of inert tracers at surface associated with the model configurations and physical processes, including vertical resolution, wet scavenging, dry deposition and sub-grid cloud mixing. In addition, with the help of these sensitivity simulations, more quantitative estimates are also provided at the process level when explaining the differences in inert tracers between WRF/CMAQ and the other models.**

*Changes in manuscript: The introduction section has been modified to clarify the research goal and the reasons that inert tracers were used. The results section is reorganized so that the impact of different physical treatment in CTMs on inert tracers are discussed one by one.*

The authors find very little transport from the lateral BC to the surface for tracer BC3 (< 250 hPa). It needs to be discussed in more detail, if this means that stratospheric ozone does not impact surface ozone in the study area at all. However, the top boundary condition would need to be considered for such an investigation. Considering only lateral influx with BC3, given that the models had only 1, 3 or 5 model levels above 250 hPa, seems not sufficient. A realistic top boundary is required to simulate the impact of upper tropospheric and stratospheric ozone on surface ozone.

**Response: First, we would like to emphasize that the BC3 results shown in this study are seasonal averages over broad regions. Therefore, they would not be expected to show the impact of episodic stratospheric intrusion events affecting specific monitors within the domain, and the relatively low seasonally and regionally averaged BC3 values do not indicate that such events did not happen. Second, neither CMAQ nor CAMx consider downward fluxes through the top of the modeling domain. Therefore, the only direct estimates of stratospheric influences within the domain are from the BC3 tracers which represent stratospheric ozone at the location of the lateral boundaries and are then advected and potentially mixed downward within the regional domain. Finally, we would like to emphasize that on a hemispheric-to-global scale, there is a large contribution of stratospheric ozone to free tropospheric ozone. Therefore, while the BC2 tracer represents free tropospheric ozone at the location of the lateral boundaries for the regional scale models, it implicitly includes stratospheric contributions occurring at larger space and time scales, and the important contribution of BC2 to surface tracer concentrations**

**across the regional domain therefore implies that the stratospheric ozone at hemispheric-to-global scale impacts surface ozone.**

*Changes in manuscript: We do not believe any changes to the manuscript are needed in response to this comment.*

A clarification of the setup of the BC tracers would be helpful. Did the values of the ozone BC from the global model agree with observations? Where the BC placed at exactly the same position for all models or did the location of the model boundaries vary from model to model? This could have had an impact on the results.

**Response: We agree that adding this information may be helpful for readers. The reviewer also made a good point that the location of the model boundaries may vary from model to model and may have had an impact on the results. WRF/CMAQ, WRF/CAMx and COSMO-CLM/CMAQ have the same (or very similar) modeling domains, making their results of inert tracers comparable to each other. However, WRF/DEHM has a very different domain coverage. Therefore, in the revised manuscript WRF/DEHM results are only used in a relative sense to assess the effects of dry deposition on simulated tracer concentrations.**

*Changes in manuscript: More details have been added in the section of "chemically inert tracers" regarding the setup of the tracers. Given that the models use different vertical resolution, Figure 2 has also been modified to better demonstrate how each model attributes lateral boundary ozone at different altitude ranges to BC1, BC2 and BC3. The lateral boundary conditions of ozone for all models are derived from C-IFS and have been evaluated against observations (Hogrefe et al., 2018). This statement is added in the section of "model description". The modeling domains and the analysis domain have also been added in the revised paper and shown are in Figure 1a.*

The main finding of the authors is that vertical mixing differs between the models. It would be interesting to elaborate on the reasons for this. The authors could provide a confirmation of this finding by studying profiles of primary species emitted over the domain. Also, a statement on which of the different mixing scheme leads to the best results would give useful information to the reader.

**Response: The differences in vertical turbulent mixing are either due to different parameterizations used in different CTMs (e.g. WRF/CMAQ and WRF/CAMx in this study), or due to the different meteorological input fields (e.g. WRF/CMAQ and COSMO-CLM/CMAQ in this study). It is true that profiles of primary species emitted over the domain may help to confirm the model differences in vertical turbulent mixing. However, the vertical profiles are not available for WRF/CAMx and WRF/DEHM as such data were not required to be archived in AQMEII3. The vertical profiles of inert tracers for WRF/CMAQ and COSMO-CLM are compared to show their potential difference in vertical mixing. Identifying the scheme that leads to the best results is beyond the scope of the paper. The goal of this analysis is to demonstrate that the process of vertical turbulent mixing is an important source of uncertainty when estimating the impact of lateral boundary ozone on the ozone levels at the surface.**

*Changes in manuscript: The reasons for different vertical turbulent mixing have been clearly mentioned when comparing each model pair. The importance of vertical mixing in causing the tracer differences has been supported by the sensitivity simulations of WRF/CMAQ for the model pair of WRF/CMAQ and*

*WRF/CAMx, and by the analysis of PBL height and vertical profiles of inert tracers for the model pair of WRF/CMAQ and COSMO-CLM/CMAQ.*

DMA8 is the only time aggregation of the discussed BC results. The choice of DMA8 for the given applications needs to be better motivated. As the focus of the study is to identify differences in vertical mixing, night-time values should not be disregarded as it is most likely the case for DMA8 of ozone. It is during night time that different vertical mixing (or the lack thereof) can have the largest impact on surface values. Likewise it would be interesting to check if the time period for the calculation of DM8A was the same for the models. Differences in the diurnal cycle can be an indication of different simulation of vertical mixing.

**Response: We agree that the choice of DM8A may not describe the complete picture of the model difference in vertical mixing. Therefore, at the revised manuscript also presents an analysis of the diurnal cycles in the model differences. As the reviewer pointed out, we do find that in some cases, the differences in the diurnal cycle can be an indication of the differences in vertical mixing between different models. The time period for calculating the DM8A was not exactly the same across the models, but the difference is usually within two hours.**

*Changes in manuscript: In the revised paper, the diurnal cycles of the model differences in the inert tracers at the surface have been added, and the results are shown as averages over sub-regions on a seasonal basis.*

The colour scale for absolute values could be improved. In my printout the magenta (e.g. 25-30 ppb range, top, Fig 3) looks very similar to the red colors indicating high values.

**Response: We apologize for the problems in the color scales.**

*Changes in manuscript: The color scales have been adjusted for easier readability.*